# Unraveling the Matrix: Proteomic Profiling Reveals Stromal ECM Dysregulation in Severe Early-Onset Fetal Growth Restriction

**DOI:** 10.3390/ijms262211179

**Published:** 2025-11-19

**Authors:** Stefano Ginocchio, Maxwell C. McCabe, Amanda R. Flockton, Diane L. Gumina, Kirk C. Hansen, Shuhan Ji, Emily J. Su

**Affiliations:** 1Division of Reproductive Sciences, Department of Obstetrics and Gynecology, University of Colorado School of Medicine, Aurora, CO 80045, USA; stefano.ginocchio@cuanschutz.edu (S.G.); amanda.flockton@cuanschutz.edu (A.R.F.); diane.gumina@gmail.com (D.L.G.); shuhanji0615@gmail.com (S.J.); 2Department of Biochemistry and Molecular Genetics, University of Colorado School of Medicine, Aurora, CO 80045, USA; kirk.hansen@cuanschutz.edu; 3Division of Maternal-Fetal Medicine, Department of Obstetrics and Gynecology, University of Colorado School of Medicine, Aurora, CO 80045, USA

**Keywords:** severe fetal growth restriction, umbilical artery Doppler velocimetry, human placenta, villous stromal fibroblasts, extracellular matrix, fibroblast cell-derived matrix

## Abstract

An underdeveloped placental vasculature is a cardinal feature in severe, early-onset fetal growth restriction with absent/reversed umbilical artery Doppler end-diastolic velocities (FGR_a/r_). Tissue microenvironment is a key mediator of angiogenesis; yet, the role of placental villous stromal extracellular matrix (ECM) in FGR_a/r_ remains unknown. We applied an ECM-optimized, proteomic workflow to villous tissue and placental fibroblast cell-derived matrices (CDM) from FGR_a/r_, gestational age-matched controls, and uncomplicated term pregnancies. No significant differences were detected in villous tissue, although there was a trend toward increased type I collagen and fibronectin in FGR_a/r_ placentas. FGR_a/r_ CDM, however, appeared distinct from both control groups, with elevated matrisome abundance, greater insolubility of matrisome-associated proteins, and 44 differentially expressed matrisome proteins. Fibronectin emerged as a central network hub among differential matrisome proteins, interacting with thrombospondin-1, vitronectin, and transglutaminase-2, all of which were enriched in FGR_a/r_ CDM, suggesting excessive deposition and crosslinking. In contrast, regulators of ECM remodeling and TGFβ activity, including fibrillin-1, decorin, and syndecan-4, were depleted. These features suggest a pro-fibrotic, dysregulated stroma with diminished remodeling capacity. Our findings establish the first, comprehensive proteomic map of human placental stromal matrisome and provide a molecular framework for understanding how aberrant ECM organization contributes to placental dysfunction.

## 1. Introduction

In severe, early-onset fetal growth restriction (FGR), the presence of absent or reversed umbilical artery Doppler end-diastolic velocities (FGR_a/r_) is indicative of an underdeveloped placental vasculature. This reduces surface area for maternal–fetal exchange and places excess strain on the fetal heart. As no preventive or therapeutic measures currently exist, delivery is required to avert stillbirth. This most often occurs in the extremely (<28 weeks) or very (28–31 weeks) preterm periods, where the severe growth restriction is further compounded by the risks of prematurity. This lack of optimal management paradigms highlights the importance of understanding mechanisms that underlie perturbed placental angiogenesis.

Tissue microenvironment, comprising extracellular matrix (ECM), various stromal cell types like fibroblasts (FBs), and signaling molecules, is a well-established mediator of various biologic processes, including angiogenesis. ECM is a three-dimensional, dynamic, and complex structure, comprising core matrisome proteins (collagens, glycoproteins, and proteoglycans) and matrisome-associated proteins (MAPs), which include ECM regulators, ECM-affiliated proteins (EAPs), and secreted factors [1,2,3]. Core matrisome proteins serve as the main contributors to the structural foundation and mechanical strength of tissue, acting as scaffolds for cells, whereas MAPs include ECM-remodeling enzymes, growth factors, and other proteins that are structurally related to ECM proteins, all of which impact biomechanical properties specific to individual tissues [2,4,5,6,7].

In addition to providing structural scaffolding for cells and tissues, the ECM also serves as a key regulator of cellular functions. FBs are the main stromal cell type responsible for ECM production, secreting fibrous proteins such as collagens and fibronectin (a glycoprotein) to generate the interstitial matrix, affecting tissue integrity, strength, and elasticity [8]. These proteins also communicate to adjacent cells through extracellular signaling domains and integrin cell surface receptors, impacting cellular adhesion, proliferation, and migration [9], all of which are essential to vascular development. The basement membrane, a specialized, sheet-like ECM structure that influences endothelial cell (EC) function, is also known to affect these same cellular processes and has been shown to be highly abundant in mouse placental tissue [8,10]. Each organ has its own distinct, tissue-specific ECM, varying in organization and complexity [10]. This ECM can become dysregulated as a result of various pathologic circumstances, further driving disease progression via its effects on adjacent cells, such as ECs.

One example of dysregulated ECM is the microenvironment of solid tumors, which displays aberrant deposition, composition, organization, and post-translational modification of the ECM as compared to that of adjacent, normal tissue [11,12]. This typically results in tumor ECM that is more abundant, denser, and stiffer, which in turn, reduces nutrient and oxygen supply, inhibits cytotoxic stress responses, induces epithelial–mesenchymal transition, and promotes angiogenesis, all of which are conducive to tumor progression [9,13].

In contrast to tumor microenvironments and cancer-associated fibroblasts, which have been extensively studied [14], there is limited knowledge surrounding human placental stromal ECM, especially in the setting of pregnancy-related pathology. Previously employed methods successfully generated baseline characterization of murine placental tissue [10] but did not utilize human tissue or include models of pregnancy dysfunction. Individual reports have demonstrated expression of ECM proteins such as fibrillin-1 (FBN1), fibulin-5, and tenascin within villous stroma [15,16,17]. Data mining of a cDNA UniGene database for *Homo sapiens* also found 102 ECM genes among approximately 10,000 mRNA species expressed in the human placenta [18]. Follow-up immunofluorescent analysis of first-trimester and term placental tissue specimens by these authors confirmed the expression of fibronectin, collagen I, collagen IV, fibrillin, thrombospondin, and tenascin within the stroma, whereas laminin was only expressed in term stroma [18]. Shear-wave elastography studies also suggest that pregnancies complicated by pathologies related to suboptimal placental function (e.g., FGR or preeclampsia) exhibit higher stiffness as determined by Young’s modulus [19,20,21,22], suggesting an in vivo correlation between placental microenvironment and function.

Mechanistically, we previously found that ECs isolated from FGR_a/r_ placentas display impaired angiogenic capacity in the form of deficient proliferative and migratory properties [23,24,25]. ECM generated from placental stromal fibroblasts—termed “cell-derived matrices (CDM)”—also had the capacity to modulate EC properties [26]. Specifically, CDM from uncomplicated, control pregnancies partially abrogated the impaired angiogenic phenotype of severe FGR ECs, whereas CDM obtained from severe FGR placentas attenuated control EC proliferation and migration. Immunofluorescence further suggested that these downstream alterations in EC behavior were mediated by aberrant deposition of specific ECM proteins, such as collagen I and fibronectin.

Taken together, these data suggest that placental stromal microenvironment—specifically villous stromal ECM that directly abuts distal placental vessels—plays a role in regulating placental angiogenesis. Given the paucity of knowledge surrounding this particular component of the placenta, our objective was to establish matrisome signatures of villous tissue and CDM obtained from three pregnancy cohorts: (1) FGR_a/r_, (2) Preterm controls (PTCs), and (3) Term controls (TCs).

## 2. Results

### 2.1. Subject Demographics

Clinical characteristics for subjects are presented in Table 1. Except for gestational age, all continuous variables were normally distributed. Although Kruskal–Wallis analysis demonstrated a significant difference (*p* < 0.0001) in gestational age at delivery overall, post hoc comparisons demonstrated no significant difference (*p* = 0.576) between PTC and FGR_a/r_ subjects despite a difference in median gestational age of approximately 5 weeks between the two cohorts. As anticipated, neonatal birthweight (*p* < 0.0001) and birthweight percentile (*p* < 0.0001) differed significantly among groups. However, on post hoc analysis, there was no significant difference in birthweight percentile between PTCs and TCs (*p* = 0.61), whereas significant differences remained when comparing TC versus FGR_a/r_ (*p* = 0.0012) and PTC versus FGR_a/r_ (*p* = 0.0002). Similarly, all six FGR_a/r_ subjects exhibited either absent or reversed umbilical artery end-diastolic velocities (*p* = 0.0002). Route of delivery also significantly differed (*p* = 0.0002), with all FGR_a/r_ and TC subjects delivered via Cesarean section in the absence of labor. All PTCs experienced preterm labor with or without intact membranes, delivering vaginally. Differences in male/female neonatal sex proportions existed between groups, but these were not statistically significant.

### 2.2. The Human Placental Matrisome

To obtain a global characterization of ECM composition within the human placenta, whole villous tissue samples underwent global proteomics via a two-fraction, ECM-optimized method, producing soluble (cellular) and insoluble ECM (iECM) fractions for independent analysis, as previously described [10,27]. This analysis resulted in 2563 protein identifications across both fractions, and the sum of the two analyzed fractions was used for the initial overall global comparison. Of these, 201 were identified as matrisome proteins, with 92 further subcategorized as core matrisome and the remaining 109 designated as MAPs.

No significant differences in abundance of all proteins or matrisome proteins were identified between TC, PTC, and FGR_a/r_ cohorts (Appendix A). Fibrinogen alpha chain (FGA), fibrinogen beta chain (FGB), and fibrinogen gamma chain (FGG) were present in disproportionately high quantities among all three cohorts, accounting for 79.1% ± 1.9% of all glycoproteins. However, we observed no significant difference in the abundance of any fibrinogen chain between the clinical groups (Appendix A). Furthermore, because of residual maternal and fetal blood within villous tissue despite multiple washes, and as fibrinogens are abundant blood plasma components often factored out of analyses of ECM components [28,29], we opted to exclude fibrinogens in our subsequent analyses. This led to proportions of core matrisome components that were more consistent with ECM composition in other tissues [10,30].

After exclusion of fibrinogens, an average of 75.5% ± 1.3% of label-free quantification (LFQ) signal intensity for matrisome proteins across both analyzed fractions was assigned to core matrisome proteins, with the remaining 24.5% ± 1.3% made up of MAPs. There were also no differences in total abundance of core matrisome proteins between TC, PTC, and FGR_a/r_ groups (Figure 1A). However, one-way ANOVA querying the proportional composition between core matrisome proteins and MAPs demonstrated a significant difference in proportion (*p* = 0.0291) overall between percent signal intensity assigned to each ECM category and clinical cohort. Post hoc comparison further identified a significant increase in the proportion of core matrisome proteins in FGR_a/r_ villous tissue as compared to PTC tissue (FGR_a/r_: 79.1%, PTC: 72.1%, TC: 74.4%; *p* = 0.0268; Figure 1B). However, there were no significant differences in the proportions of any individual matrisome subcategory, including collagens, glycoproteins, proteoglycans, ECM regulators, EAPs, and secreted factors among TC, PTC, and FGR_a/r_ specimens (Figure 1C).

A primary benefit of the two-fraction extraction performed in this study is that it allows for assessment of protein solubility by calculating the percentage of total signal identified in the iECM fraction for each protein. The iECM fraction is primarily composed of highly abundant proteins that can be covalently crosslinked, providing structure, support, and organization to tissues and impacting tissue characteristics such as architecture, rigidity, and density [4,31]. ECM protein solubility is largely determined by these post-translational modifications, or association of soluble factors to iECM proteins, and has been shown to change with age and various pathological states [32,33,34]. An average of 96.7% ± 0.3% of core matrisome protein signal was identified in the insoluble fraction, whereas only 38.1% ± 2.1% of MAP signal was detected in the iECM. There were no significant differences in overall percent signal identified in the iECM fraction of core matrisome proteins between TC, PTC, and FGR_a/r_ cohorts (Figure 1D), whereas FGR_a/r_ specimens exhibited MAPs that were significantly more insoluble than TC (overall one-way ANOVA *p* = 0.0441, TC vs. FGR_a/r_ *p* = 0.0386; Figure 1E). Proportionally, there were also no iECM differences among the three groups in any matrisome subcategory (Figure 1F).

Given the various cell types in villous tissue and our overall goal of establishing placental stromal ECM signatures, we opted to focus on insoluble core matrisome proteins and MAPs. Of the 201 iECM matrisome proteins identified in the iECM fraction, those with zero or values close to the detection limit in 50% or more samples were further excluded, yielding 128 proteins for subsequent analysis. After normalization by the total LFQ signal intensity, statistical analyses comparing iECM matrisome proteins between TC, PTC, and FGR_a/r_ villous tissue demonstrated no significant differentially expressed proteins. Furthermore, principal component analysis (PCA) did not reveal distinct clustering of cohorts (Appendix A). This was not entirely surprising, given the inherent variability in sampling of human specimens and heterogeneous cellular composition of villous tissue, all of which contribute to variability in ECM production, remodeling, and degradation.

To identify proteome changes with significant effects on the overall biomechanical and structural properties of the matrix, we opted to focus on high-abundance structural ECM proteins within the iECM fraction. To do this, we performed a partial least squares discriminant analysis (PLS-DA), a multivariate technique that does not assume normally distributed variables, allowing it to maximize covariance between predictors and class labels [35]. Log transformation, while improving the normality of left-skewed data, also compresses the differences among high-abundance proteins and amplifies the differences among low-abundance proteins [36]. Thus, we opted to utilize normalized but non-transformed data for PLS-DA analysis, ensuring that the analysis was sensitive to differences in proteins most likely to influence matrix function. PLS-DA also confirmed overlap among the three cohorts (Figure 2A). Despite this, the variable importance in projection (VIP) scores calculated in the PLS-DA model identified some ECM proteins that could contribute to differences in TC, PTC, and FGR_a/r_ matrisome signatures. VIP, which quantitatively estimates the ability of each variable to discriminate between cohorts, is typically considered significant when ≥1 [37,38,39,40]. However, we opted to focus on ECM proteins with VIP ≥ 2, as mathematical models suggest using thresholds higher than 1 when the number of dependent variables queried is fewer than 200 and/or when none demonstrate a significant FDR-corrected *p*-value [41]. VIP scoring identified three proteins—the a1 (COL1A1) and a2 (COL1A2) chains of type I collagen, and fibronectin 1 (FN1)—that could play a sizeable role in differentiating the three clinical cohorts (Figure 2B). The a1 chain of type II collagen (COL2A1), which assembles in homotrimers to form collagen II, also had a VIP score suggesting that its relative abundance could also predict segregation of TC, PTC, and FGR_a/r_ villous tissue.

Although each of these matrisome proteins did not individually reach the threshold for statistical significance, all were identified as key discriminative proteins in the multivariate PLS-DA model, with higher relative abundance of COL1A1, COL1A2, and FN1 and lower relative abundance of COL2A1 in FGR_a/r_ villous stroma. Furthermore, type I collagen and fibronectin (COL1A1, COL1A2, FN1) were the most abundantly expressed collagen and glycoprotein, respectively, among all samples, accounting for an average of 44.0% ± 1.2% of all core matrisome signal within the iECM fraction. We detected a trend whereby the summed signal for COL1A1, COL1A2, and FN1 was proportionally higher in FGR_a/r_ villous tissue as compared to either TC or PTC specimens (TC: 42.0%, PTC: 42.5%; FGR_a/r_: 47.6%; *p* = 0.1061; Figure 2C). Importantly, while blood/plasma contamination could contribute to overall FN1 abundance, total fibrinogen levels showed low correlation with FN1 abundance, indicating that differences in fibronectin observed here were not driven by varying levels of blood contamination. Type I collagen, typically composed of two a1 chains and one a2 chain, is the primary structural element of most tissues, contributing significantly to tissue stiffness and often elevated in fibrotic states [42]. Together with FN1, which provides nucleation sites for collagen fibrillogenesis and forms a provisional ECM scaffold [43,44,45], these findings suggest that a synergistic effect between type I collagen and fibronectin may be an important characteristic in FGR_a/r_ villous stroma.

In total, our data suggest that the FGR_a/r_ placental matrisome may display unique structural features, and even small fold-changes in highly abundant proteins reflect comparatively large absolute differences in abundance, potentially impacting stromal ECM properties. However, limitations in using whole villous tissue to establish placental stromal ECM signatures exist, given the diversity of cell types, which each produce their own ECM and/or alter stromal ECM content and architecture. These diverse contributors to whole tissue composition make it possible that our villous tissue findings are diluted by signal from other non-stromal constituents.

### 2.3. Matrisome of Placental FB-Derived CDM

To more clearly delineate specific features of stromal ECM between FGR_a/r_, PTC, and TC placental specimens, we also examined CDM generated from stromal FBs isolated from these three cohorts. In the intervening period between analysis of villous tissue and that of CDM, a new mass spectrometer (Bruker timsTOF SCP) was acquired, allowing for significantly improved sensitivity and data quality for analysis of CDM samples. This analysis resulted in greater overall protein identifications (7242 across both analyzed fractions), including 313 matrisome proteins, with 112 comprising the core matrisome and 201 categorized as MAPs.

In confirmation that fibrinogen chains in villous tissue were most likely seen due to residual blood plasma, all three fibrinogen chains (FGA, FGB, and FGG) were sparsely identified or undetectable in all CDM samples. Similar to villous tissue, total peak intensities of all proteins were similar between TC, PTC, and FGR_a/r_ CDM samples (Appendix A). In contrast to villous tissue, the total matrisome protein abundance was significantly higher in the FGR_a/r_ cohort as compared to either PTC or TC (overall one-way ANOVA *p* = 0.0007; FGR_a/r_ vs. TC *p* = 0.0144; FGR_a/r_ vs. PTC = 0.0006; Figure 3A). This finding persisted when looking solely at MAP abundance (overall one-way ANOVA *p* = 0.0002, FGR_a/r_ vs. TC: *p* = 0.0033; FGR_a/r_ vs. PTC: *p* = 0.0002; Figure 3B). Within the core matrisome, significant differences among groups remained (overall one-way ANOVA *p* = 0.0280), but post hoc comparisons only showed significantly higher signal intensities in FGR_a/r_ as compared to PTC specimens (FGR_a/r_ vs. PTC *p* = 0.0226; Figure 3C).

For CDM, core matrisome proteins made up an average of 35.0% ± 1.0% of detected matrisome protein signal, whereas MAPs accounted for 65.0% ± 1.0% (Figure 3D). No significant differences were found on two-way ANOVA when comparing these proportions between TC, PTC, and FGR_a/r_. Similarly, there were no significant differences in the proportional abundance of collagens, glycoproteins, proteoglycans, ECM regulators, EAPs, or secreted factors among the three groups (Figure 3E).

Similar to villous tissue, core matrisome proteins within CDM were more insoluble than MAPs. In this case, an average of 70.2% ± 1.5% of total core matrisome protein signal was detected in the insoluble fraction, compared to 46.5% ± 2.3% for MAPs. No significant differences in overall percent insolubility of core matrisome proteins existed among the three cohorts (Figure 3F). However, MAPs within FGR_a/r_ specimens demonstrated significantly more insolubility (FGR_a/r_: 56.4% ± 1.3%) than those from either TC (46.8% ± 3.1%) or PTC (46.5% ± 1.2%) (overall one-way ANOVA *p* < 0.0001, TC vs. FGR_a/r_ *p* = 0.0094, PTC vs. FGR_a/r_ *p* = 0.0057; Figure 3G) groups. The iECM proportions of collagen, glycoprotein, proteoglycan, ECM regulator, EAP, and secreted factor signal were also similar between all three groups (Figure 3H).

As in villous tissue, we opted to focus on protein intensity identified in the iECM fraction for subsequent CDM analysis to better reveal alterations in matrix composition between cohorts. Matrisome identifications within the iECM fraction (313 total) were then filtered to remove proteins that were undetectable in 50% or more samples, resulting in 267 matrisome proteins for subsequent proteomic analysis. In contrast to villous tissue, CDM displayed 61 proteins that were significantly different among TC, PTC, and FGR_a/r_ FB-generated matrix (Appendix A). Unsupervised PCA showed distinct clustering of the FGR_a/r_ cohort, with TC and PTC groups exhibiting more similarity to one another than to FGR_a/r_, despite minor overlap in 95th percentile confidence regions (Figure 4A). This was further supported by unbiased hierarchical clustering as depicted by the heatmap of the top 50 distinguishing proteins between groups by ANOVA (Figure 4B).

Of the 61 iECM matrisome proteins that significantly differed among the three cohorts, 44 of these showed directional differences confined to FGR_a/r_ as compared to both TCs and PTCs (Table 2; Appendix A). Eight of these exhibited significantly higher signal intensities in FGR_a/r_ than in TCs and PTCs, whereas the other 36 showed significantly lower signal intensities than the two control cohorts. The remaining proteins either had significant differences between all three groups, differences only between PTC and TC subjects, or had only one control group that differed from FGR_a/r_.

FN1 was the most abundantly expressed protein in the iECM fraction across all CDM specimens. Similar to the trend seen in villous tissue, FN1 signal intensity was significantly higher in FGR_a/r_ CDM as compared to CDM generated from either TC or PTC FBs (*p* = 0.0006; Figure 5A). Protein–protein interactions were queried using the STRING database and the STRINGApp plugin in Cytoscape (v3.10.3) [46,47,48], focusing solely on iECM proteins that differed significantly when comparing FGR_a/r_ to both TC and PTCs. Full functional network analysis identified FN1 as a main hub, connecting with 26 of the 44 proteins that differed between FGR_a/r_ and both control groups within the iECM (Appendix A).

Physical network analysis, which spotlights direct, physical interactions between proteins, showed especially strong confidence in FN1 interactions with thrombospondin-1 (THBS1), transglutaminase-2 (TGM2), vitronectin (VTN), fibulin-1 (FBN1), syndecan-4 (SDC4), fibroblast growth factor-2 (FGF2), and decorin (DCN) (Figure 6A). THBS1, TGM2, and VTN were all increased in FGR_a/r_ CDM as compared to both TC and PTC CDM (*p* < 0.01 for both; Figure 5A). THBS1, a core matrisome glycoprotein, promotes ECM stabilization, activation of latent transforming growth factor-beta (TGFb), and inhibits ECM proteolysis [49,50,51]. The specific effects of excess VTN, also a core matrisome glycoprotein, on ECM properties are less clear, although high levels of either VTN or THBS1 have been associated with impairment in angiogenic properties [51,52,53,54,55]. TGM2 is an enzyme that crosslinks fibronectin, collagens, and other ECM proteins [56], and thus, the net effect of elevated levels of THBS1, VTN, and TGM2, along with their direct interaction with increased FN1, suggests excessive ECM deposition and crosslinking, which could result in a stiffened matrix.

In contrast, FBN1, SDC4, FGF2, and DCN were all decreased in FGR_a/r_ CDM compared to both TC and PTC CDM (*p* < 0.05 for these four; Figure 5B). The glycoprotein FBN, which serves as a scaffold for ECM components, and the proteoglycan DCN, which is required for collagen fibril alignment, can both act to sequester TGFb, regulating its activity within the matrix [57,58,59]. FGF2 inhibits FB activation and, along with SDC4, upregulates enzymes involved in ECM remodeling [60,61,62]. Together, decreased FBN1, DCN, FGF2, and SDC4 further suggest an ECM that is more resistant to remodeling and one that demonstrates a more pro-fibrotic signature.

Gene ontology (GO) enrichment analysis for biological processes of all proteins that differed in FGR_a/r_ CDM, as compared to both control groups, identified several overrepresented functional categories (Figure 6B). These included expected ones, such as “ECM organization”, “cell adhesion”, and “integrin-mediated signaling pathway”. Notably, GO also demonstrated that “tube morphogenesis,” “regulation of cell migration,” and “vasculature development” were among the top ten detected biological processes, corroborating the importance of the FGR_a/r_ microenvironment on placental vasculature.

## 3. Discussion

In this study, we investigated the human placental matrisome signature in subject-matched, villous tissue and placental FB-generated CDM by leveraging a two-fraction, ECM-optimized proteomic pipeline developed across several previous studies [27,63]. We found that, while there were no significant differences in placental villous tissue ECM content between TC, PTC, and FGR_a/r_ specimens, use of a stringent VIP score suggested that elevated abundance of COL1A1 and COL1A2—the alpha chains that form type I collagen—and FN1 in FGR_a/r_ villous tissue may play a role in the overall matrisome signature of FGR_a/r_ placentas. Collagen I and fibronectin are the two most abundant ECM proteins in human villous tissue among all three cohorts, together accounting for more than 30% of all detected matrisome signals. Thus, even in the absence of significant differences in signal intensities between cohorts, small changes in highly abundant proteins could substantially affect stoichiometry, protein–protein interaction, or integrin–ECM complex formation, all of which impact tissue biology [64,65]. For instance, it is well-established that an organized, fibrillar fibronectin matrix is necessary for the maintenance of type I collagen and several other ECM proteins, such as THBS1 [66,67,68]. More specifically, the quantity and architecture of fibronectin fibers impact collagen binding sites and nucleation of collagen fibrillogenesis [45]. From a cellular perspective, an in vitro model of human umbilical vein endothelial cells seeded onto hydrogels coated with varying ratios of collagen I and fibronectin found that even modest changes in ratios significantly altered cell tractional forces and shape [69], which further regulate ECM homeostasis. Together, these data suggest that even small, relative alterations in highly abundant ECM proteins, like those identified here between villous tissue groups, could substantially impact stromal characteristics.

Using placental FB-generated CDM to focus more specifically on ECM stromal signatures, we found that, although there were no proportional differences in matrisome subcategories between the three cohorts, FGR_a/r_ CDM exhibited significantly higher total matrisome protein abundance than either TC or PTC CDM. Additionally, MAPs from FGR_a/r_ CDM were significantly more insoluble than those from either control cohort, suggesting greater integration of these proteins into the insoluble matrix. Unbiased proteomic analysis also demonstrated clustering of FGR_a/r_ CDM specimens, with 44 iECM proteins showing significant differences between FGR_a/r_ and both control cohorts. Thirty-six of these iECM proteins were lower in FGR_a/r_ CDM, including several regulators of ECM remodeling. Tenascin-X (TNXB), the most significantly different iECM protein between groups, was lower in FGR_a/r_ CDM. Low levels of TNXB have been implicated in disorganized collagen fibrils and in classical-like Ehlers–Danlos syndrome, likely through its role in regulating collagen fibrillogenesis and fibril spacing [70,71,72,73]. Thus, lower TNXB abundance in FGR_a/r_ CDM aligns with the generally more dysregulated ECM phenotype observed in FGR_a/r_ samples in this study, supported by reductions in other collagen fibrillogenesis-regulating proteins (DCN, FBN, etc.). TNXB-deficient mice have also been found to exhibit decreased vascular density, suggesting that low TNXB itself could also negatively regulate angiogenesis [72,73].

Other structural ECM proteins were also seen with less signal intensity in FGR_a/r_ CDM, including FBN1 and certain FACIT collagens, which are critical for structural support and organization of the ECM but are generally not highly abundant in fibrotic tissue [74]. The activity of TGFb, which promotes FB activation and a pro-fibrotic phenotype, is additionally regulated by many ECM proteins. FBN1 and DCN, proteins known to temper TGFb activity [75], were seen in lower quantities in FGR_a/r_ CDM, potentially allowing for TGFb overactivation. In conjunction with elevated FN1, THBS1, and TGM2, FGR_a/r_ CDM displays a signature of excessive crosslinking potential, resistance to ECM remodeling, and dysregulated matrix deposition and organization. In total, findings from both villous tissue and CDM support that the FGR_a/r_ stromal signature appears more fibrotic and simultaneously more susceptible to dysregulated collagen fibrillogenesis and improper remodeling.

The available, but limited, human data also support the concept of a compromised FGR_a/r_ placental microenvironment. For instance, shear-wave elastography studies have demonstrated that placentas compromised by conditions like FGR are stiffer, which is supported by other studies that reported stromal fibrosis and more myofibroblast activation specifically in FGR_a/r_ placentas as compared to those from GA-matched, appropriately grown pregnancies [19,20,21,22,76,77]. Notably, these differences persisted even when comparing FGR_a/r_ placentas to FGR placentas with abnormal but *preserved* (as opposed to absent or reversed) umbilical artery end-diastolic Doppler velocities [76,77]. Another group also demonstrated that in severe, early-onset preeclampsia, a condition with placental pathology that often overlaps with FGR_a/r_, PE placentas exhibited more fibrosis-related proteins such as collagens, fibronectin, and α-smooth muscle actin in vivo and in vitro, with preeclamptic stromal FBs being more sensitive to TGFb1 stimulation [78]. These previous findings further support the pro-fibrotic phenotype identified here in the FGR_a/r_ placenta.

ECM homeostasis depends primarily on four central factors: (1) Rate of ECM production, (2) Rate of ECM degradation, (3) Biophysical and biochemical properties of individual ECM proteins, and (4) Effect of ECM on cellular constituents and their reciprocal effects on matrix [79]. Disruption of any of these factors, whether through excessive deposition, inadequate remodeling, or altered architecture, can lead to pathology, which has been demonstrated in numerous examples. For instance, keloid scar formation arises from a disorganized, fibroproliferative collagen response [80,81]. Others have found that in skin, aging increases intracellular fibronectin in FBs but impaired secretion, disorganized fibril formation, and enhanced ECM degradation [82,83]. On the other hand, aging impacts vessels differently, where secondary to enhanced crosslinking, collagens develop as longer and thicker fibrillar structures [84]. Post-translational changes in other proteins, such as fibronectin, also increase ECM stiffness. Within this aging model, angiogenesis is impaired [84]. In contrast, stiffening of the ECM promotes tumor angiogenesis [85,86].

Our current data suggest that placental stromal ECM within FGR_a/r_ placentas exhibits a signature suggestive of increased stiffness and disorganization as a result of excessive deposition of provisional ECM proteins, overactive crosslinking, and impaired capacity for degradation and remodeling. On first glance, this appears to contradict prior findings in our lab, as we previously found that FGR_a/r_ CDM exhibited less fluorescent intensity on immunofluorescent imaging for type I collagen and fibronectin as compared to TC CDM [26]. However, FGR_a/r_ CDM also displayed aberrant, disorganized architecture, with several areas appearing “contracted,” leading to areas on the image without any notable fluorescence, as compared to a uniform-appearing matrix with even collagen I and fibronectin staining in control CDM. It is possible that type I collagen and fibronectin were both present throughout the field of image in FGR_a/r_ CDM, and the disrupted topography limited epitope availability for antibody binding, underestimating the total amount of collagen or fibronectin. Alternatively, erratic and disorganized ECM deposition by FGR_a/r_ FBs is also possible, resulting in areas of abnormally contracted and dense collagen and fibronectin fibrils that exceed the maximum fluorescent signal, also resulting in underestimation of total collagen or fibronectin within these regions. Notably, TC placental endothelial cells showed impaired angiogenic properties when plated on FGR_a/r_ CDM, further supporting the idea that excessive crosslinking and aberrant remodeling, as suggested by our proteomic data, may limit the interpretation of immunofluorescent imaging quantification of collagen and fibronectin in FGR_a/r_ CDM. This CDM phenotype is also more consistent with that of aged vessels, where angiogenesis is also faulty [84]. Alternatively, impaired angiogenesis in FGR_a/r_ placentas could also be explained by the altered basement membrane composition observed in our MS data, with decreased laminin content and increased COL4A2, both of which comprise the basement membrane that, in vivo, is in direct contact with placental endothelial cells.

This highlights one limitation of the current study: we interrogated villous tissue and CDM at a single time point, yet ECM homeostasis exists in a non-static responsive equilibrium that adapts to developmental, physiological, and pathological cues. Additionally, our CDM model lacked other cell types that could impact the placental microenvironment, including Hofbauer cells (placental macrophages), ECs, and trophoblasts. Another potential limitation is the concern that neither TCs nor PTCs are ideal controls. Placentas obtained from TCs are at a substantially more mature gestational age than those within our FGRa/r cohort, and both ECM homeostasis and angiogenesis can vary throughout gestation. Similarly, placentas from PTCs, although appropriately grown for gestational age, are delivered preterm for some other pathologic etiology, such as preterm labor or preterm premature rupture of membranes, which can lead to other potential confounders (e.g., increased inflammation). Our PTC subjects all underwent labor and delivered vaginally, which subjected placentas to intermittent, physiologic hypoxia caused by uterine contractions. More recent data also demonstrate that, in these spontaneous preterm deliveries of appropriately grown fetuses, some placentas display pathologic findings indicative of placental dysfunction, which is near-universal in FGR_a/r_ [87,88,89,90,91]. Despite limitations of our control populations, proteomic analysis suggested that matrices derived from TC and PTC fibroblasts were more similar to each other than they were to those from FGR_a/r_ subjects. If placental dysfunction indeed underlies spontaneous preterm births, this may indicate that the matrisome of PTC placentas may have the capacity to adapt to a suboptimal environment, which could preserve fetal growth. Alternatively, the similarities between TC and PTC CDM could suggest that matrisome alterations are not central to the pathophysiology of spontaneous preterm birth. It is also possible that significant differences were limited only to CDM because a newer, more sensitive mass spectrometer became available to us only for CDM analysis, which occurred several months after the MS query of villous tissue. Lastly, we acknowledge that our sample size of six subjects per group has the power to detect only large effect sizes, and thus, there may have been other biologically significant findings that were not identified in this current analysis. However, this suggests that the significant differences detected in this study are likely ones that yield biologically meaningful effects.

In summary, our study demonstrates differences in FGR_a/r_ placental ECM composition as compared to both TC and PTC specimens, with a signature involving overproduction of key structural ECM proteins, altered basement membrane composition, elevated crosslinking enzyme abundance, and diminished capacity for matrix remodeling. The importance of the tissue microenvironment in regulating biologic function is clear in other organs, including the liver, heart, lung, and several solid tumors. Given the dynamic nature of tissue stromal matrix and the increasing emergence of therapies aimed at modifying ECM properties, understanding alterations to the placental microenvironment in pathologies such as FGR_a/r_ and their contributions to placental dysfunction is crucial for identifying novel targets for prevention or treatment. The work presented here represents a significant step toward addressing this critical gap in our knowledge of the molecular mechanisms underlying placental dysfunction. In the future, increasing sample size may also allow us to explore correlations between matrisome parameters and clinical characteristics such as birthweight, placental weight, and birthweight and placental weight percentile. It will also be important to understand mechanisms underlying these matrisome changes, including pathways such as TGFb-mediated myofibroblast activation, impact of post-translational modifications such as crosslinking and its effects on growth factor sequestration, as well as regulation of stromal cell-mediated ECM remodeling.

## 4. Materials and Methods

### 4.1. Subjects

After approval by the Colorado Multiple Institutional Review Board (COMIRB), subjects from the three cohorts were identified: (1) Singleton pregnancies complicated by severe, early-onset FGR with an estimated fetal weight of less than the 10th percentile for gestational age AND absent or reversed end-diastolic umbilical artery velocities (FGR_a/r_), (2) Gestational age-matched, singleton pregnancies that resulted in spontaneous preterm deliveries of appropriately grown fetuses (PTCs), and (3) Uncomplicated, full-term pregnancies also with appropriately grown fetuses (TCs).

Gestational age was confirmed using criteria defined by the American College of Obstetricians and Gynecologists (ACOG) [92]. Exclusion criteria for all cohorts included diabetes, history of thrombosis, antiphospholipid antibody syndrome, maternal or fetal infection, fetal anomaly, or aneuploidy. Any medical comorbidities, such as hypertensive diseases of pregnancy that could impact placental function, were also considered exclusion criteria for the control cohorts. Eligible subjects were then approached, and informed consent was obtained.

After delivery, actual birth weight was used to confirm growth restriction in the FGR_a/r_ cohort and appropriate growth (10th to 90th percentile for gestational age) in both control groups. PTCs that exhibited placental pathology suggestive of maternal or fetal vascular malperfusion as per the Amsterdam Placental Workshop Consensus Statement [93] were excluded.

### 4.2. Villous Tissue Sampling

Within 60 min after delivery, a 1 × 1 cm piece of full-thickness tissue was obtained from the parenchyma midway between the umbilical cord insertion site and the peripheral edge of the placenta, avoiding areas of overt infarction or abnormalities. The chorionic plate and maternal decidua were dissected off. Villous tissue was washed with PBS (Thermo Fisher Scientific, Waltham, MA, USA) five times to remove as much blood as possible, snap-frozen in liquid N_2_, and then brought to the Proteomics Core.

### 4.3. Primary Placental Villous Stromal FB Isolation

Villous stromal FBs were isolated as previously described [26]. Briefly, immediately following villous tissue sampling, 30 g of chorionic villi were dissected after removal of the basal and chorionic plates, washed in Hank’s balanced salt solution (MilliporeSigma; Darmstadt, Germany) with gentamicin (Thermo Fisher Scientfic) and once with RPMI-1640 (Thermo Fisher Scientfic) with 2% FBS and 1% antibiotic/antimycotic solution (C-RPMI-1640). Samples were minced and digested in C-RPMI-1640 with 0.28% Collagenase D (Roche, Basel, Switzerland), 0.25% Dispase II (MilliporeSigma), 0.002% DNase I (MilliporeSigma) for 2 h in a shaking incubator at 37 °C. A discontinuous Percoll (MilliporeSigma) gradient of ten layers (10, 20, 30, 35, 40, 45, 50, 55, 60, and 70%) was prepared during the digestion period in two 30-mL glass centrifuge tubes. After digestion, the cell suspension was filtered and strained, washed, centrifuged, and resuspended in 6 mL C-RPMI-1640, with 3 mL of the suspension carefully layered on each Percoll gradient. Gradients were then centrifuged at 3060 rpm for 20 min at 22 °C without brakes. Percoll layers three to five (located between 30 and 40%) containing cells were then transferred to a fresh tube, washed with 30 mL C-RPMI-1640, and centrifuged again. This cell pellet was then collected and resuspended in 20 mL of Endothelial Growth Media-2 (Lonza) and centrifuged at 1500 rpm × 5 min at 22 °C.

Cells were then counted and seeded at a density of 1 × 10^7^ cells per dish onto 100-mm dishes pre-coated with Attachment Factor solution (Cell Applications). Cells were cultured for an additional 10 days, followed by trypsinization, which effectively eliminates trophoblast cells. Cells within this suspension then underwent selection with CD31 Dynabeads (Thermo Fisher Scientfic), thereby removing microvascular ECs, and then plated once again. The remaining FBs were then cultured in full Fibroblast Growth media-2 (FGM-2; Lonza, Walkersville, MD, USA) supplemented with 2% FBS, insulin, fibroblast growth factor, and gentamicin/amphotericin B. The cellular morphology of FBs, which are distinct from trophoblast and ECs, was confirmed via bright-field microscopy, and cell purity was assessed via immunofluorescence and flow cytometry.

### 4.4. Generation of Placental Villous Stromal FB CDMs

As previously described [26,94], CDMs were generated from TC, PTC, and FGR_a/r_ stromal FBs from the same subjects where villous tissue was obtained. In this case, FBs were cultured on 100-mm tissue culture plates that had undergone coating with 0.2% (*w*/*v*) gelatin (BD) for 1 h at 37 °C. Plates were then washed, crosslinked with 1% (*v*/*v*) glutaraldehyde (Sigma, Darmstadt, Germany), washed once again, followed by quenching with 1 M glycine (VWR). Thereafter, plates were washed and then incubated in full FGM-2 for 1 h at 37 °C.

After preparation of gelatin-coated plates, isolated FBs were seeded in full FGM-2 at a density of 3.5 × 10^6^ cells/100 mm plate with the goal of confluence within 12–16 h. After reaching 100% confluence, the medium was replaced with full FGM-2 supplemented with 50 mg/mL freshly prepared ascorbic acid. This medium was replaced daily with freshly prepared ascorbic acid. Once a uniform matrix was visualized via bright-field microscopy, FBs were washed with PBS and then brought to the Proteomics Core for further processing. CDM generated from each subject was replicated at least in duplicate, with the exception of one subject’s FBs, which were not viable.

### 4.5. Sample Preparation for Liquid Chromatography–Tandem Mass Spectrometry (LC-MS/MS)

#### 4.5.1. Villous Tissue

Villous tissue was prepared using previously published ECM-optimized proteomic methods [27,63]. Tissues snap-frozen in liquid nitrogen were lyophilized overnight using a FreeZone 4.5 L benchtop freeze dryer (Labconco #7750020, Kansas City, MO, USA) and milled to a fine powder prior to extraction. Three milligrams of material from each sample was combined with 100 mg of 1 mm glass beads (Next Advance #GB10) and homogenized in 200 μL/mg of decellularization buffer (50 mM Tris-HCl (pH 7.4), 0.25% CHAPS, 25 mM EDTA, 3 M NaCl) supplemented with 1X Halt Protease Inhibitor Cocktail (Thermo Fisher Scientific #78429) at power 8 for 3 min (Bullet Blender, Model BBX24, Next Advance, Inc., Troy, NY, USA). Homogenate was then vortexed (power 8) for 20 min at 4 °C, spun at 18,000× *g* (4 °C) for 15 min, and the supernatant was collected. Decellularization was repeated for a total of three washes, homogenizing and vortexing samples before each collection, and all washes were pooled to generate the cellular fraction. Pellets were then treated with freshly prepared hydroxylamine buffer (1 M NH_2_OH·HCl, 4.5 M Gnd·HCl, 0.2 M K_2_CO_3_, pH adjusted to 9.0 with NaOH) at 200 μL/mg of the starting tissue dry weight. Samples were homogenized at power 8 for 1 min and incubated at 45 °C with shaking (1000 rpm) for 4 h. Following incubation, the samples were spun for 15 min at 18,000× *g*, and the supernatant was removed before being stored at −80 °C as the ECM fraction. All fractions were subsequently subjected to overnight enzymatic digestion with trypsin (1:100 enzyme/protein ratio) using a filter-aided sample preparation approach as previously described [95]. Digested peptides were desalted using Pierce™ C18 spin tips (Thermo Fisher Scientific #84850) according to the manufacturer’s protocol, and the resulting peptide solution was quantified using the Pierce™ quantitative colorimetric peptide assay (Thermo Fisher Scientific #23275).

#### 4.5.2. CDM

Subject-matched CDM samples were prepared using previously published ECM-optimized proteomic methods [27,63] adapted for this sample type. Each PBS-washed CDM plate was washed with 1 mL decellularization buffer [50 mM Tris-HCl (pH 7.4), 0.25% CHAPS, 25 mM EDTA, 3 M NaCl] supplemented with 1X Halt Protease Inhibitor Cocktail (Thermo Fisher Scientific #78429). After buffer addition, each plate was agitated at 75 RPM using an orbital shaker (Thermo Fisher Scientific #SK4000) at 4 °C for 2 h. Supernatant was then removed from each plate by tilting the plate and carefully pipetting the supernatant from the side of the plate before being reserved as the cellular fraction. Each plate was then washed with 1 mL 1X PBS to remove residual cellular components by agitating briefly at 75 RPM on an orbital shaker before carefully removing and discarding the supernatant. The remaining matrix on each plate was then extracted by adding 1 mL of 6 M guanidine hydrochloride, 100 mM ammonium bicarbonate, and agitating overnight at 75 RPM on an orbital shaker at RT (25 °C). After incubation, the surface of each plate was thoroughly scraped using a cell scraper (Celltreat #229310, Ayer, MA, USA) to collect residual ECM. The supernatant was carefully removed from the side of the plate and stored as the ECM fraction. All fractions were subsequently subjected to overnight enzymatic digestion with trypsin (1:100 enzyme/protein ratio) using a filter-aided sample preparation approach as previously described^3^. Digested peptides were desalted using Pierce C18 spin tips (Thermo Fisher Scientific #84850) according to the manufacturer’s protocol, and the resulting peptide solution was quantified using the Pierce^TM^ quantitative colorimetric peptide assay (Thermo Scientific Fisher #23275).

### 4.6. LC-MS/MS Analysis

#### 4.6.1. Villous Tissue

Global proteomics of villous tissue was carried out on a Fusion Lumos Tribrid mass spectrometer (Thermo Fisher Scientific) coupled to an EASY-nLC 1200 (Thermo Fisher Scientific) through a nanoelectrospray LC-MS interface. Eight µL of each sample containing 2 µg of protein was injected into a 20 μL loop using the autosampler. The analytical column was then switched online at 400 nL/min over an in-house-made 100 μm i.d. × 150 mm fused silica capillary packed with 2.7 μm CORTECS C18 resin (Waters; Milford, MA, USA). LC mobile phase solvents consisted of 0.1% formic acid (FA) in water (buffer A) and 0.1% FA in 80% acetonitrile (buffer B, Optima LC/MS, Fisher Scientific, Pittsburgh, PA, USA). After 22 μL of sample loading at a maximum column pressure of 700 bar, each sample was separated on a 120 min gradient at a constant flow rate of 400 nL/min. The separation gradient for cell fractions consisted of 6% buffer B from 0 to 3 min, followed by a linear gradient from 6 to 42% buffer B from 3 to 105 min, while a linear gradient from 6 to 24% buffer B was utilized from 3 to 105 min for the ECM fraction. Gradient elution was followed by a linear increase to 55% buffer B from 105 to 110 min and further to 95% buffer B from 110 to 111 min. Flow at 95% buffer B was maintained from 111 to 120 min to remove the remaining peptides. Data acquisition was performed using the instrument supplied Xcalibur (version 4.5) software operating in positive ion mode. Survey scans of *m*/*z* 375–1600 were followed by higher energy collisional dissociation (HCD) MS/MS (30% collision energy) using the standard automatic gain control target and a 35 ms maximum injection time with an isolation width of 1.6 *m*/*z*. The Orbitrap was used for MS1 and MS2 detection at resolutions of 120,000 and 50,000, respectively. Dynamic exclusion was performed after fragmenting a precursor one time for a duration of 45 s. Singly charged ions were excluded from HCD selection.

#### 4.6.2. CDM

Desalted peptides were adjusted to a protein concentration of 25 ng/µL with 0.1% FA in preparation for MS analysis. Digested peptides were loaded into autosampler vials and analyzed directly using a NanoElute liquid chromatography system (Bruker, Germany) coupled with a timsTOF SCP mass spectrometer (Bruker, Germany). Peptides were separated on a 75 µm i.d. × 15 cm separation column packed with 1.9 µm C18 beads (Bruker, Germany) over a 90-min elution gradient. Buffer A was 0.1% FA in water, and buffer B was 0.1% FA in acetonitrile. Instrument control and data acquisition were performed using Compass Hystar (version 6.0) with the timsTOF SCP operating in parallel accumulation-serial fragmentation (PASEF) mode under the following settings: mass range 100–1700 *m*/*z*, 1/k/0 Start 0.7 V s cm^−2^ End 1.3 V s cm^−2^; ramp accumulation times were 166 ms; capillary voltage was 4500 V, dry gas 8.0 L min^−1^ and dry temp 200 °C. The PASEF settings were: five MS/MS scans (total cycle time, 1.03 s); charge range 0–5; active exclusion for 0.2 min; scheduling target intensity 20,000; intensity threshold 500; collision-induced dissociation energy 10 eV.

### 4.7. MS Data Processing

Data from both villous tissue and CDM samples were searched using MSFragger v3.8 via FragPipe v 20.0 [96]. Precursor tolerance was set to ±60 ppm, and fragment tolerance was set to ±0.25 Da for preliminary searches, after which data were recalibrated within the MSFragger pipeline and searched using precursor and fragment tolerances of ±20 ppm. Data was searched against SwissProt, restricted to *Homo sapiens*, with added common contaminant sequences [97] (20,410 total sequences). Enzyme cleavage was set to semi-specific trypsin for all samples. Fixed modifications were set as carbamidomethyl (C). Variable modifications were set as oxidation (M), oxidation (P) (hydroxyproline), Gln->pyro-Glu (N-term Q), and acetyl (Protein N-terminus). Label-free quantification was performed using IonQuant v1.9.8 with match-between-runs enabled and default parameters. Results were filtered to 1% false discovery rate (FDR) at the peptide and protein level. Extracted fractions were searched separately and merged after database searching. Reported core matrisome (ECM proteins) and matrisome-associated proteins were annotated with MatrisomeDB [30].

The Search Tool for the Retrieval of Interacting Genes/Proteins (STRING) tool was used to further examine protein–protein interactions of differentially expressed insoluble ECM (iECM) proteins within CDM via the STRINGApp plugin in Cytoscape (v.3.10.3). Interactions with confidence > 0.4 are depicted [98], and disconnected nodes were excluded to simplify network visualization. The resulting network was visualized using the yFiles Organic layout. Functional enrichment analysis, including Gene Ontology enrichment analysis, was also performed in STRING. Enrichment terms with an FDR < 0.01 were considered significant.

### 4.8. Statistical Analysis

Six subjects per group yielded 80% power (a = 0.05) to detect a Cohen’s f effect size of 0.55. Sample replicates for each subject were averaged, and numerical data are reported as means of these averages ± standard error of the mean (SEM). Data were analyzed using GraphPad Prism 10 (GraphPad Software Inc., La Jolla, CA, USA). The Shapiro–Wilk test was used to determine whether data were normally distributed. Thereafter, clinical demographics and ECM characteristics were compared among the three cohorts using one-way ANOVA for parametric data, with Tukey’s multiple comparison post hoc testing, when appropriate. For non-Gaussian data, Kruskal–Wallis tests were applied. Proportionality was compared using two-way ANOVAs, and if applicable, followed also by Tukey’s post hoc testing. A *p*-value of *p* < 0.05 was considered significant.

After pre-processing to ensure quality, proteomic data with peak intensity values were uploaded to Metaboanalyst 6.0, where the data were normalized to sum and log-transformed, followed by one-factor ANOVA to detect proteins with statistically significant differences among the three clinical cohorts. A FDR correction of 0.05 was applied for multiple testing errors.

## 5. Conclusion

This comprehensive, proteomic map of the human placental stromal matrisome suggest a pro-fibrotic, dysregulated villous stroma in FGR_a/r_ placentas, providing a molecular framework for understanding how aberrant EC organization contributes to placental function. Elucidation of mechanisms underlying how EC-ECM interactions impact placental angiogenesis in severe, early-onset FGR will help inform development of potential interventions.

## Figures and Tables

**Figure 1 ijms-26-11179-f001:**
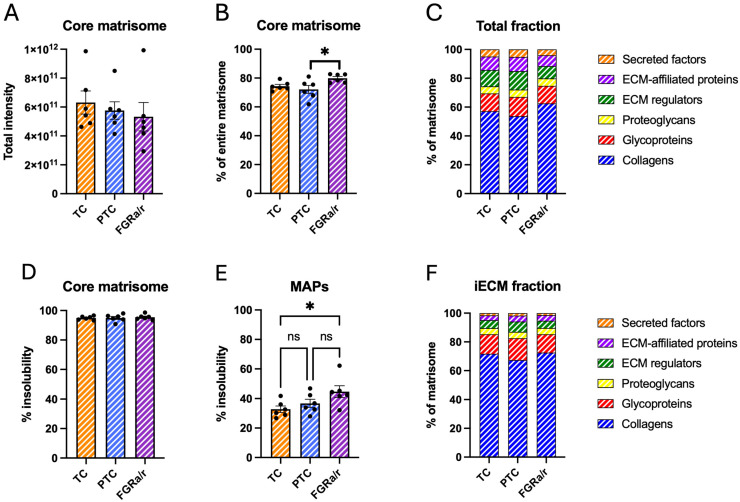
Villous matrisome global characteristics. (**A**) One-way ANOVA demonstrates similar total intensities of core matrisome between TC, PTC, and FGR_a/r_ cohorts. (**B**) The matrisome is comprised of two main categories: core matrisome proteins (collagens, glycoproteins, and proteoglycans) and MAPs (ECM regulators, ECM-affiliated proteins, secreted factors). Plot depicts the percentage of total matrisome signal assigned to core matrisome proteins (overall one-way ANOVA; *p* = 0.0291). Tukey’s post hoc comparison also showed a significant difference in proportion of core matrisome proteins in FGR_a/r_ villous tissue only as compared to PTC tissue (* *p* = 0.0268). (**C**) Two-way ANOVA showed no significant differences in proportional composition of the six matrisome subcategories among the three groups across both analyzed fractions. (**D**) Core matrisome protein insolubility, displayed as the percentage of total protein signal identified in the iECM fraction, was similar between cohorts by one-way ANOVA. (**E**) MAP insolubility significantly differed among the three groups based on one-way ANOVA (*p* = 0.0441), although the only post hoc significant difference occurred between FGR_a/r_ and TC (* *p* = 0.0386). (**F**) Within the iECM fraction, proportional composition of matrisome subcategories based on detected intensity within the iECM fraction was similar across TC, PTC, and FGR_a/r_ as determined via two-way ANOVA. ((**A**–**F**): *n* = 6/cohort; ns: non-significant).

**Figure 2 ijms-26-11179-f002:**
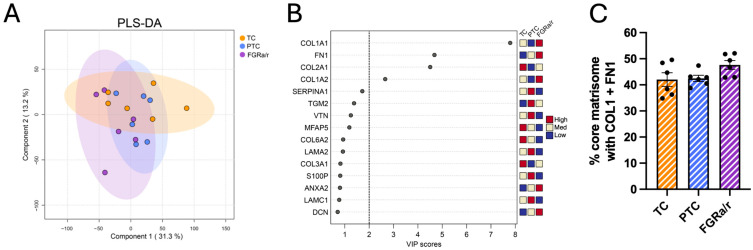
Proteomic analysis of villous iECM matrisome. (**A**) Supervised PLS-DA clustering analysis of the iECM fraction from villous tissue demonstrates substantial overlap between TC, PTC, and FGR_a/r_ villous tissue proteomic profiles. (**B**) VIP analysis using a cutoff of VIP > 2 suggested elevated COL1A1, COL1A2, and FN1 in FGR_a/r_ villous tissue. (**C**) Percent of total core matrisome intensity assigned to COL1A1, COL1A2, and FN1 further suggested a trend toward more insoluble collagen I and fibronectin in FGR_a/r_ placentas, as determined by one-way ANOVA (*p* = 0.1061). ((**A**–**C**): N = 6/cohort).

**Figure 3 ijms-26-11179-f003:**
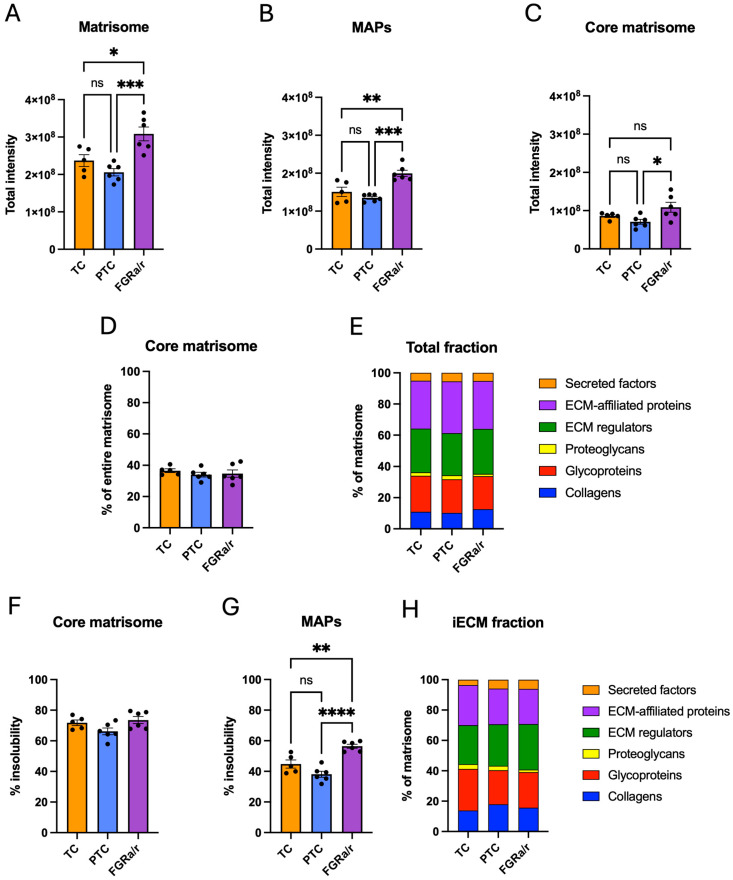
CDM global matrisome characteristics. (**A**) Total matrisome protein intensity significantly differed among the three cohorts (overall one-way ANOVA *p* = 0.0007) and was significantly higher in the FGR_a/r_ cohort as compared to TC (* *p* = 0.0144) and PTC (*** *p* = 0.0006) CDM based on Tukey multiple comparison tests. (**B**) MAPs showed significant differences by one-way ANOVA overall (*p* = 0.0002), with significantly higher peak intensities in FGR_a/r_ as compared to TC (** *p* = 0.0033) and PTC (*** *p* = 0.0002) CDM. (**C**) Although core matrisome proteins also significantly differed between groups by one-way ANOVA (*p* = 0.0280), only FGR_a/r_ and PTC CDM exhibited significant differences (* *p* = 0.0226). (**D**) Proportional composition of CDM matrisome by category is depicted based on total detected intensity, with all three cohorts displaying similar proportions of core matrisome proteins. (**E**) No significant differences between groups based on two-way ANOVA were found in the proportional composition of the six matrisome subcategories across both analyzed fractions. (**F**) Core matrisome protein insolubility, displayed as the percentage of total protein signal identified in the iECM fraction, was similar between cohorts. (**G**) MAP insolubility differed significantly as detected by one-way ANOVA between all three groups (*p* < 0.0001), with FGR_a/r_ exhibiting more insoluble MAPs than TC (** *p* = 0.0030) and PTC (**** *p* < 0.0001) CDM. (**H**) Proportions of matrisome subcategory intensity within the iECM fraction remained similar across TC, PTC, and FGR_a/r_ CDM via two-way ANOVA. ((**A**–**H**): TC: *n*= 5, PTC: *n* = 6; FGR_a/r_: *n* = 6; ns: non-significant).

**Figure 4 ijms-26-11179-f004:**
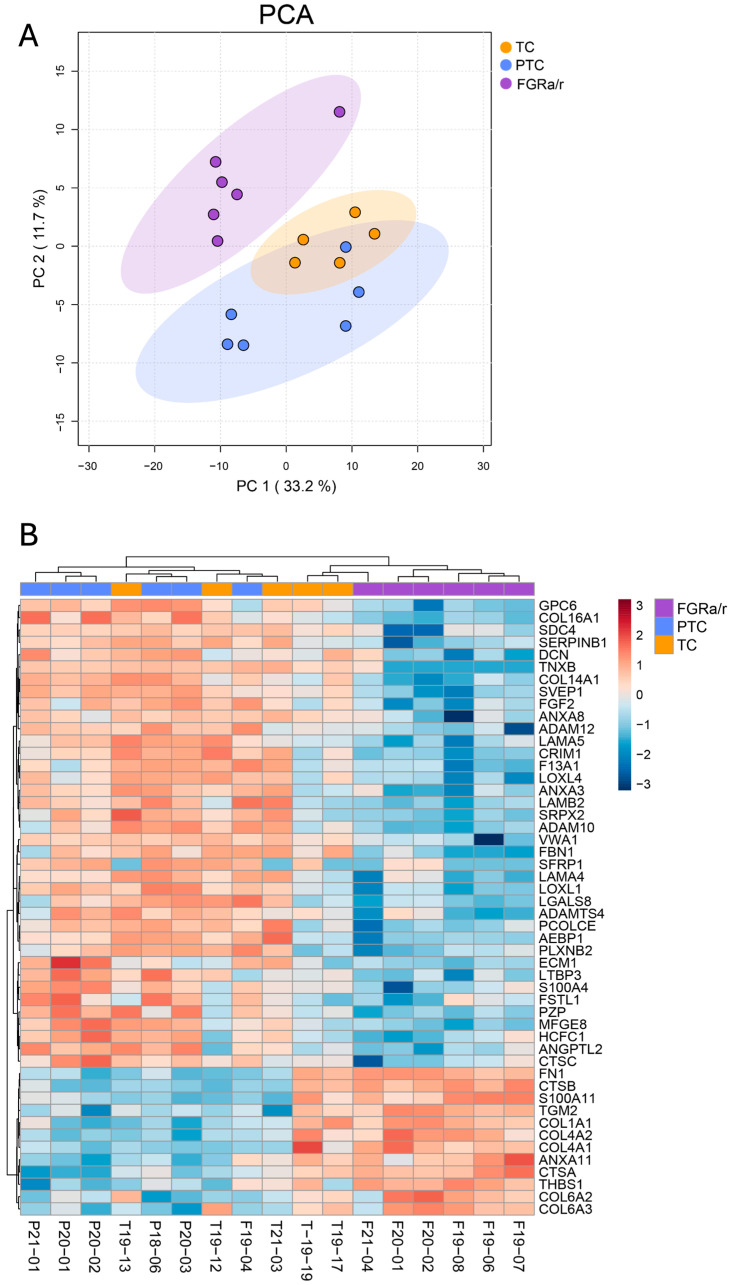
Proteomic analysis of CDM iECM matrisome. (**A**) Unsupervised clustering via PCA showed separation of FGR_a/r_ CDM from other groups, with TC and PTC appearing more similar to each other than to FGR_a/r_. (**B**) Unbiased hierarchical clustering depicting the top 50 proteins by ANOVA significance also showed distinct clustering of FGR_a/r_ CDM. (**A**,**B**) TC, *n* = 5; PTC, *n* = 6; FGR_a/r_, *n* = 6.

**Figure 5 ijms-26-11179-f005:**
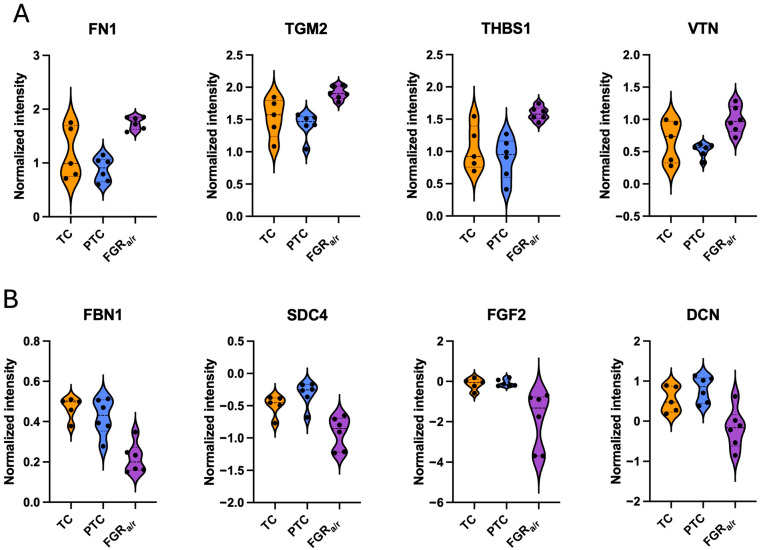
Box plots of normalized iECM concentrations for select matrisome proteins. (**A**) As compared to TC and PTC CDM, FGR_a/r_ CDM exhibited significantly higher signal intensities for FN1 (*p* = 0.0092), THBS1 (*p* = 0.0144), and TGM2 (*p* = 0.0197), and VTN (*p* = 0.0395) and (**B**) lower signal intensities of FBN1 (*p* = 0.0066), FGF2 (*p* = 0.0092), DCN (*p* = 0.0179), and SDC4 (*p* = 0.0232). Normalized intensity represents acquired signal intensity after sum normalization and log-transformation. (ANOVA FDR cutoff of 0.05; (**A**,**B**): TC: *n* = 5, PTC: *n* = 6; FGR_a/r_: *n* = 6).

**Figure 6 ijms-26-11179-f006:**
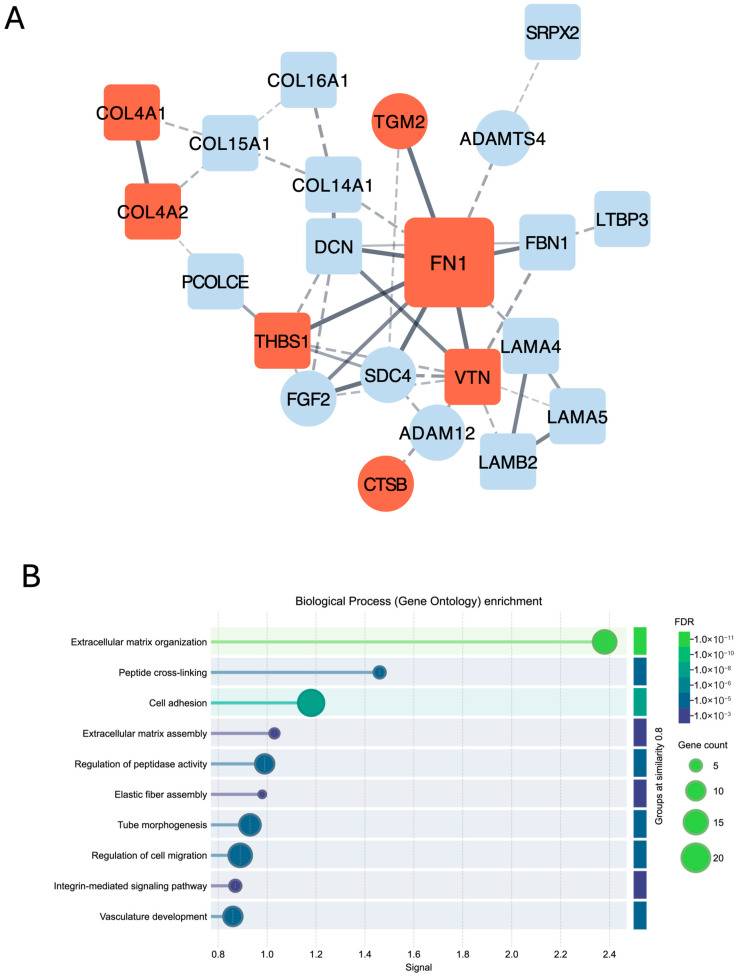
STRING and Cytoscape physical network analysis. This network analysis utilized CDM matrisome proteins that differed significantly and with the same directionality in FGR_a/r_ as compared to both TC and PTCs in the iECM fraction. Matrisome proteins that were increased in FGR_a/r_ are in red nodes, whereas those that were decreased in FGR_a/r_ are in blue nodes. Circular symbols denote core matrisome proteins, while diamonds denote MAPs. (**A**) Physical network analysis identified FN1 as a main hub and depicts physical interactions between proteins, with especially strong confidence in FN1 interactions with THBS1, TGM2, SDC4, and FBN1. (**B**) GO enrichment lists the top ten biological processes associated with the full network analysis of these iECM proteins.

**Table 1 ijms-26-11179-t001:** Clinical demographics.

	TC (*n* = 6)	PTC (*n* = 6)	FGR_a/r_ (*n* = 6)
Maternal age (years)	34 ± 1.4	29 ± 2.7	33 ± 3.2
Gestational age at delivery (weeks) *	39.0 ± 0.07	32.4 ± 0.7	28.0 ± 1.5
Doppler velocimetry (absent/reversed) ^#^	0/6	0/6	4/2
Neonatal birthweight (g) ^	3359 ± 157	1931 ± 129	602 ± 60
Neonatal birthweight percentile ^@^	53.2 ± 11.4	63.7 ± 6.9	4.3 ± 1.2
Placental weight (g) ^&^	520 ± 30	288 ± 23	152 ± 33
Neonatal sex (M/F)	4/2	3/3	5/6
Route of delivery (vaginal/C-section) ^#^	0/6	6/6	0/6

* *p* < 0.0001 by Kruskal–Wallis (TC vs. FGR_a/r_: *p* = 0.0010; PTC vs. FGR_a/r_: *p* = 0.5762); ^#^ *p* < 0.001 by Chi-squared test; ^ *p* < 0.0001 by one-way ANOVA (TC vs. FGR_a/r_: *p* = 0.0010; PTC vs. FGR_a/r_: *p* = 0.0001); ^@^ *p* = 0.0004 by one-way ANOVA (TC vs. PTC: *p* = 0.6100; TC vs. FGR_a/r_: *p* = 0.0012; PTC vs. FGR_a/r_: *p* = 0.0020); ^&^ *p* < 0.0001 by one-way ANOVA (TC vs. PTC: *p* = 0.0001; TC vs. FGR_a/r_: *p* < 0.0001; PTC vs. FGR_a/r_: *p* = 0.0012).

**Table 2 ijms-26-11179-t002:** Differentially expressed iECM proteins.

**iECM Proteins Significantly Higher in FGR_a/r_ as Compared to Both TC and PTC**			
**Gene Symbol**	**Protein**	**Category**	**FDR *p*-Value**
CTSB	Cathepsin B	ECM regulator	0.0081124
FN1	Fibronectin	Glycoprotein	0.0091698
COL4A2	Collagen IV a2	Network-forming collagen	0.0097667
THBS1	Thrombospondin-1	Glycoprotein	0.01437
TGM2	Transglutaminase-2	ECM regulator	0.01972
S100A11	Protein S100-A11	Secreted factor	0.01972
COL4A1	Collagen IV a1	Network-forming collagen	0.023061
VTN	Vitronectin	Glycoprotein	0.039511
**iECM Proteins Significantly Lower in FGR_a/r_ as Compared to Both TC and PTC**			
**Gene Symbol**	**Protein**	**Category**	**FDR *p*-Value**
TNXB	Tenascin-X	Glycoprotein	0.00080041
SVEP1	Sushi, von Willebrand factor type A, EGF, and pentraxin domain-containing protein 1	Glycoprotein	0.0015087 0.0015087
COL14A1	Collagen XIV a1	FACIT collagen	0.0059602
FBN1	Fibrillin-1	Glycoprotein	0.0065528
COL16A1	Collagen XVI a1	FACIT collagen	0.0065528
CRIM1	Cysteine-rich motor neuron 1 protein	Glycoprotein	0.0070293
GPC6	Glypican-6	EAP	0.0081931
PZP	Pregnancy zone protein	ECM regulator	0.0081931
ADAM10	Disintegrin and metalloproteinase domain-containing protein 10	ECM regulator	0.0091698
LAMB2	Laminin b2	Glycoprotein	0.0091698
FGF2	Fibroblast growth factor 2	Secreted factor	0.0091698
LOXL4	Lysyl oxidase homolog 4	ECM regulator	0.0091698
ADAM12	Disintegrin and metalloproteinase domain-containing protein 12	ECM regulator	0.0091698
SERPINB1	Leukocyte elastase inhibitor	ECM regulator	0.009169
AEBP1	Adipocyte enhancer-binding protein 1	Glycoprotein	0.0097667
SRPX2	Sushi repeat-containing protein 2	Glycoprotein	0.0097667
LTBP3	Latent transforming growth factor beta-binding protein 3	Glycoprotein	0.01527
PCOLCE	Procollagen C-endopeptidase enhancer 1	Glycoprotein	0.01527
LAMA5	Laminin a5	Glycoprotein	0.015623
ANXA3	Annexin A3	EAP	0.015711
F13A1	Coagulation factor XIII A chain	ECM regulator	0.017934
DCN	Decorin	Proteoglycan	0.017934
VWA1	Von Willebrand Factor 1	Glycoprotein	0.019643
CTSC	Dipeptidyl peptidase 1	ECM regulator	0.01972
LAMA4	Laminin a4	Glycoprotein	0.01972
ADAMTS4	A disintegrin and metalloproteinase with thrombospondin motifs 4	ECM regulator	0.021216
SDC4	Syndecan-4	EAP	0.023202
LGALS8	Galectin-8	EAP	0.023202
PLXNB2	Plexin B2	EAP	0.032235
ANXA8	Annexin A8	EAP	0.033602
PAMR1	Inactive serine protease PAMR1	ECM regulator	0.033804
CTHRC1	Collagen triple helix repeat-containing protein 1	Glycoprotein	0.037121
THSD4	Thrombospondin type 1-domain-containing protein 4	Glycoprotein	0.043657
SEMA3B	Semaphorin-3B	EAP	0.043985
COL15A1	Collagen XVa1	Multiplexin collagen	0.045474
CRELD1	Protein disulfide isomerase CRELD1	Glycoprotein	0.047579

## Data Availability

MS proteomic data for villous tissue and for CDM have been deposited to the ProteomeXchange Consortium via the PRIDE partner repository under identifier PXD070632.

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
