# Peer review of "Unraveling the Matrix: Proteomic Profiling Reveals Stromal ECM Dysregulation in Severe Early-Onset Fetal Growth Restriction"

_ijms, 2025, doi:10.3390/ijms262211179_

Round 1
Reviewer 1 Report
Comments and Suggestions for Authors
This paper aims to characterize the matrisome of placental samples from early-onset fetal growth restriction (FGR) cases, gestational age–matched (preterm) controls, and term controls.
Studying FGR is crucial, as it is a common pregnancy complication associated with immediate and lifelong consequences for the child. Moreover, little is known about its underlying pathogenesis beyond the established role of placental dysfunction. This study employs mass spectrometry–based profiling of the extracellular matrix (ECM) in placentas affected by severe early-onset FGR, revealing patterns of dysregulation and their potential implications for placental dysfunction.
I only have minor suggestions for improvement, summarized below:
- Each legend should clearly state the statistical analyses performed and indicate the sample size. These details are sometimes difficult to discern, even when individual data points are shown.
- Supplementary Figures 1, 2, and 4 were not properly viewable on my computer (black backgrounds made the graphs barely visible). In addition, their legends lack details on data analysis methods. Without this information, it is difficult to fully assess the robustness of the findings described in the paper.
- The manuscript uses several non-standard abbreviations (e.g., CDM, FGRa/r, PTC, TC, iECM, MAP). Although these are defined, their frequency makes the paper challenging to read. Consider whether all are necessary.
- Excluding fibrinogens from subsequent analyses is reasonable. However, given that some maternal and fetal blood contamination remains, could fibrinogen levels be used to correct for variable contamination across sample groups?
- Line 182: Instead of “predict differentiation,” which may be misinterpreted by developmental biologists, consider using “predict segregation.”
- Line 190 / Figure 2C: The data are not statistically significant and show overlapping values between groups. I recommend removing the corresponding sentence and revising the paragraph to avoid overinterpretation of non-significant results. The same applies to the first paragraph of the Discussion.
- Based on my interpretation, the data suggest minimal, if any, changes in the placental villous matrisome between preterm and FGR samples. This is, in itself, very interesting and could indicate adaptive mechanisms that preserve the matrisome (perhaps due to its essential role), or that matrisome alterations are not central to the pathophysiology of adverse outcomes such as preterm birth and FGR.
- Was the sample size based on power calculations? Is the study sufficiently powered? Given that placental and birth weights are continuous variables, exploring correlations between these and matrisome parameters would strengthen the analysis. Discussion of sample size limitations and the importance of expanding the cohort would be valuable. The authors also mention that iECM analyses were performed using a more sensitive mass spectrometer- this should be clearly stated and discussed.
- Figure 4: The lack of separation between the PTC and TC groups suggests that the PTC samples may not represent pathological cases - which is good an perhaps may need to be mentioned in the results (like in the discussion).
- Additional analyses / future directions: It would have been valuable to complement the proteomic data with localization studies of key matrisome-associated proteins e.g., FN1, THBS1, and assessments of pathway activation e.g., TGF-β signalling, in the same placental samples. Including these analyses, or at least noting them as future directions, would enhance the biological interpretation of the findings and their relevance to placental morphology and cellular composition.
Reviewer 2 Report
Comments and Suggestions for Authors
The MS entitled “Unraveling the matrix: Proteomic profiling reveals stromal ECM dysregulation in severe early-onset fetal growth restriction” provides a detailed proteomic analysis of matrisome signatures in impaired placental angiogenesis. The MS improves our understanding of the dysregulation of placental development on a molecular level, that provides new routes for molecular diagnostics and therapy. The MS is quite fine and should be recommended for publication after resolving some minor issues.
1) The volcano plots of pairwise comparisons (all the quantified proteins) between three pregnancy cohorts: (1) FGRa/r, (2) Preterm controls (PTCs), and (3) Term controls (TCs) in fold change/p-values coordinates. The top hits should be provided (10 or more up- and down-regulated) and unambiguously labeled.
2) Does the scale bar of the Fig. 2B heatmap have only 3 values (-1, 0, and 1), or does it have a continuous scale? Consider indicating it more clearly.
3) Fig 5. Please provide unambiguously what is normalized intensity and put it into the figure legend.
4) While the described differences in major ECM proteins is statistically significant the difference in their level is not so high. Are there any other hits (maybe less abundant proteins) that changed more drastically? Please highlight them in the Results section and discuss their potential role in the Discussion.
5) I was unable to find your raw data. Please, provide the working URL or other link instead of “submission reference 1-20251001-034017-2204759”
